# Lymphatic Drainage Mapping with Indirect Lymphography for Canine Mammary Tumors

**DOI:** 10.3390/ani11041115

**Published:** 2021-04-13

**Authors:** Francesco Collivignarelli, Roberto Tamburro, Giovanni Aste, Ilaria Falerno, Francesca Del Signore, Francesco Simeoni, Michail Patsikas, Jacopo Gianfelici, Rossella Terragni, Valeria Attorri, Augusto Carluccio, Massimo Vignoli

**Affiliations:** 1Faculty of Veterinary Medicine, University of Teramo, 64100 Teramo, Italy; fcollivignarelli@unite.it (F.C.); ifalerno@unite.it (I.F.); fdelsignore@unite.it (F.D.S.); fsimeoni@unite.it (F.S.); acarluccio@unite.it (A.C.); mvignoli@unite.it (M.V.); 2School of Veterinary Medicine, Aristotle University of Thessaloniki, 54124 Thessaloniki, Greece; patsikm@vet.auth.gr; 3Clinica Veterinaria Val Musone, 60027 Osimo (AN), Italy; jacopogianf@libero.it; 4Clinica Veterinaria Pet Care, 40133 Bologna, Italy; terragni.rossella@gmail.com; 5Clinica Veterinaria Riviera Vet, 63066 Grottammare (AP), Italy; valeria.attorri@gmail.com

**Keywords:** sentinel lymph node, mammary gland tumors, indirect lymphography

## Abstract

**Simple Summary:**

Mammary gland tumours are the most common canine neoplasms. They account for 25–50% of all tumors diagnosed in bitches. Metastases and recurrences develop in about 35–70% of bitches following excision. The presence of regional lymph node metastases is a relevant factor affecting prognosis and treatment in cases of mammary gland tumors. The sentinel lymph node (SLN) is the first lymph node (or nodes) in the regional lymphatic basin that receives lymphatic flow from the primary neoplasm. As mammary gland tumors mainly spread by lymphatic vessels invasion, conceptually, distant metastasis should not be present if the SLN does not have evidence of a tumor burden. In the present study, an indirect lymphography was used for sentinel lymph node mapping in a cohort of 14 adult female dogs with 24 mammary tumors involving the third to the fifth mammary gland. Contrast was injected around the mammary tumor, and lymph nodes that picked up the contrast were visible after 24 h. The results of this study indicate that the lymph drainage pattern of the neoplastic mammary gland may be different for each tumor. The knowledge of the SLN of the neoplastic mammary glands should be of high importance for the surgeon not only for performing the most adequate surgical excision but also for determining an accurate post-surgical prognosis.

**Abstract:**

Mammary gland tumors are the most common canine neoplasms. They account for 25–50% of all tumors diagnosed in bitches. Metastases and recurrences develop in about 35–70% of bitches following excision. The presence of regional lymph node metastases is a relevant factor affecting prognosis and treatment in cases of mammary gland tumors. The sentinel lymph node (SLN) is the first lymph node (or nodes) in the regional lymphatic basin that receives lymphatic flow from the primary neoplasm. The aim of this study is to investigate the SLN with indirect lymphography for a mammary tumor in dogs. The knowledge of the precise drainage pattern and SLN of the neoplastic mammary glands would provide clinically relevant information to the surgeon and to the oncologist, and it would be of high importance for the surgeon not only for performing the most adequate surgical excision but also for determining an accurate post-surgical prognosis.

## 1. Introduction

Mammary gland tumors (MGT) are the most common canine neoplasms [1,2]. They account for 25–50% of all tumors diagnosed [3,4,5,6,7], and it is generally accepted that about 50% of mammary tumors are malignant [1,6,7].

The major form of treatment for canine mammary neoplasia is surgical [5,6]. In benign tumors, their accurate excision has a good prognosis, while in malignant cases, local recurrences or metastases to regional lymph nodes and distant internal organs are often observed, resulting in death [5,6,7,8,9,10]. Metastases and recurrences develop in about 35–70% of bitches following the excision of malignant epithelial tumors [5,6,7,8,9,10].

The presence of regional lymph node metastases is a relevant factor affecting prognosis and treatment in cases of breast cancer in women [11,12]. 

Opinions on the prognostic value of the neoplastic involvement of regional lymph nodes in bitches are conflicting [10,13,14]; recently, much attention has been paid to the presence of lymph node micrometastases, that is, the foci of neoplastic cells have diameters ranging from 0.2 to 2 mm, and to their relevance for prognosis in mammary neoplastic diseases in bitches [15].

Micrometastasis refers to the systemic spread of a small number of tumor cells, too few to be detected by screening tests, and is thought to be largely responsible for metastatic disease in patients originally diagnosed as node negative [16]. 

The clinical stage is important for determining the extent of the disease in animals with malignant cancers, and the status of the lymph node will help to determine whether adjuvant treatment is required [17,18].

Historically, the regional anatomic lymph node has been sampled to determine the presence or absence of metastatic disease, but there is increasing evidence that the regional anatomic lymph node is not often the sentinel lymph node [19]. Thus, because of the possibility of aberrant lymphatic drainage from the tumor, the regional anatomic lymph node may not be the first draining lymph node (or sentinel lymph node (SLN)) [19].

The SLN is the first lymph node (or nodes) in the regional lymphatic basin that receives lymphatic flow from the primary neoplasm [20,21]. Single afferent lymphatic vessels can drain into a single lymph node, multiple afferent lymphatic vessels can drain into a single lymph node, and single divergent or multiple afferent lymphatic vessels can drain into more than one lymph node [21]. 

The SLN concept is based on the theory that the metastatic process occurs in an orderly progression within the lymphatic system with tumor cells draining into a specific lymph node in a regional lymphatic field before draining into other regional lymph nodes [18,22], and has an important role as a filter and barrier for disseminating tumor cells [22]. 

A fatal outcome of canine mammary neoplasia is virtually always due to metastasis of the primary tumor to distant organs [23]. Neoplastic spread mainly occurs by lymphatic invasion [24], less commonly by hematogenous dissemination [25]. 

Conceptually, distant metastasis should not be present if the SLN does not have evidence of a tumor burden, but distant metastasis is possible if the SLN has histologic evidence of metastatic tumor cells. Hence, the status of the SLN may reflect the status of the entire regional lymphatic bed; the probability that a non-SLN is positive for metastatic disease, when the SLN is free of any tumors, is less than 0.1% [22,26].

In 2012, one study showed that, in a population of healthy dogs, intradermal or submucosal injection of iodized oil allowed for the identification of a local lymph node suspected to be the SLN of the area. Using radiography, lymph node enhancement was observed 24 h after the injection [27]. 

Another report established the feasibility of a protocol where solid tumors will have SLN mapping; the results showed that indirect lymphography allowed for the identification of at least one draining node assumed to be the SLN in 96.6% of the cases 24 h after peritumoral injection using conventional radiography or computerized tomodensitometry [20].

It is postulated that mammary neoplasia may alter the lymphatic drainage pattern forming new draining channels and recruiting a large number of lymph nodes [28,29,30]. Consequently, the knowledge of the lymph drainage of the neoplastic mammary glands should be of high importance for the surgeon not only for performing the most adequate surgical excision but also for determining an accurate post-surgical prognosis.

In veterinary clinical practice, well-established guidelines for SLN evaluation are lacking.

The aim of this study is to investigate the SLN with indirect lymphography for the mammary tumor of the dog. The knowledge of the precise drainage pattern and SLN of the neoplastic mammary glands would provide clinically relevant information to the surgeon and to the oncologist. This would be of high importance for the surgeon not only for performing the most adequate surgical excision but also for determining an accurate post-surgical prognosis.

## 2. Materials and Methods

This study was approved by the Regional Committee of Animal Research and Ethics (protocol n. 10/2020), according to the Italian law decree N. 26/2014 and to EU directive 2010/63.

Sentinel lymph node mapping with indirect lymphography was investigated in a cohort of adult female dogs.


Bitches of various breeds bearing mammary tumors with no clinical or radio graphic evidence of distant metastases were included in this study. 

Each dog underwent a clinical oncological consultation, hematobiochemical investigations and urine work, chest radiographs and abdominal ultrasounds or whole-body CT scans for staging were done. All the mammary glands were carefully evaluated for each dog, and once the mammary tumors were clinically diagnosed, the contrast study for SLN was performed. Thoracic and abdominal radiographs were performed just before and 24 h after the contrast injection to identify the site and number of the opacified nodes. In the case of uncooperative patients, sedation was provided.

Each radiograph was evaluated by a board-certified radiologist (MV). 

Lipiodol Ultra-Fluid TM (iodized ethyl-esters of the fatty acids of poppy seed oil, Guerbet, Aulnay-sous-bois, France; 480 mg iodine mL^−1^) was used. A subcutaneous contrast injection was performed with a 25 G needle into the four quadrants encircling the tumor, at the border of the neoplasm. A slow rate infiltration, 0.2–0.4 mL for each quadrant for a total of 0.8–1.6 ml of contrast medium over 1–2 min, depending on the tumor size, was administered as described previously [27]. Care was taken to avoid any contrast medium injection entering into a blood vessel or into an intratumoral fluid filled cavity. Following the injection, the area was mildly massaged to facilitate the contrast medium absorption from intramammary lymphatics.

Tumor surgical excision and SLN excision were performed. The histopathology of the nodes and mammary gland excised was performed after the surgery.

## 3. Results

The presence of peritumoral contrast was observed in all cases. Lymph nodes that picked up the contrast were visible in radiographs taken 24 h after the contrast injection.

The histopathological type of each MGT and the lymph nodes into which the efferent mammary lymphatics were drained are summarized in Table 1.

A total of 14 dogs with 24 MGT matched the inclusion criteria, there were 11 entire dogs: 2 mixed breed, 2 Cocker Spaniel, 1 Maremmano Shepherd dog, 1 Staffordshire Terrier, 1 Labrador Retriever, 1 Maltese dog, 1 Yorkshire Terrier, 1 Pinscher, 1 Pitt Bull Terrier; and 3 neutered: 1 German Shepard, 1 Epagneul Breton and 1 English Setter. The median body weight was 16.2 kg and ranged from 3 to 36 kg. The median age was 9.8 years old (ranging from 4 to 13 yr). No sedation was performed for the contrast injection (Table 1). 

### 3.1. MGT Localization 

The numbers of identified MGT were 5 in the third, 7 in the fourth, and 12 in the fifth. No tumors were detected in the first and second mammary gland in our cases (Table 1).

### 3.2. Lymph Node Drainage

The third MGT (cranial abdominal), drained into the ipsilateral axillary node in three out of five mammary tumors; one of five drained into the contralateral axillary lymph node (Figure 1); one of five drained ipsilateral accessory axillary and axillary lymph nodes were opacified (Figure 2). 

The fourth MGT (caudal-abdominal) drained into the ipsilateral superficial inguinal lymph node in six out of seven cases. In one of seven dogs, the ipsilateral superficial inguinal and ipsilateral medial iliac lymph nodes were opacified (Figure 3). 

The fifth MGT (inguinal) drained into the ipsilateral superficial inguinal lymph nodes in 8 out of 12 examined mammary tumors. In 3 out of 12 dogs the ipsilateral superficial inguinal and ipsilateral medial iliac lymph nodes were opacified. In 1 case out of 12 MGT no evidence of node contrast enhancement was observed.

Histopathology of MGT and lymph nodes are summarized in Table 1.

## 4. Discussion 

The lymphatic system is considered a main route of metastases of canine mammary cancer, while surgery of mammary tumors implies removing the tumor with the glands associated with lymphatic drainage along with the lymph nodes [19]. The SLN is defined as the hypothetical first lymph node to drain a specific body area. Consequently, if an area is affected by a tumor, and because tumor staging implies systematic nodal assessment, the SLN should be the first one to show metastases and therefore should be the lymph node that is assessed as a priority [20].

The first, or cranial thoracic, and second, or caudal thoracic, neoplastic mammary tumors usually drain into the ipsilateral axillary lymph nodes and rarely into the ipsilateral axillary and sternal lymph nodes, simultaneously [27]. The third neoplastic mammary gland usually drains into the ipsilateral axillary and superficial inguinal lymph nodes simultaneously, but sometimes only cranially into the ipsilateral axillary lymph nodes. Rarely, it drains only caudally into the ipsilateral superficial inguinal and medial iliac lymph nodes, simultaneously. The fourth, or caudal abdominal, neoplastic mammary gland usually drains only caudally into the ipsilateral superficial inguinal lymph nodes. Rarely, it drains into the ipsilateral axillary and superficial inguinal lymph nodes simultaneously. The fifth, or inguinal, neoplastic mammary gland usually drains into the ipsilateral superficial inguinal lymph nodes but rarely does it also drain into the ipsilateral popliteal lymph node and into a lymphatic plexus at the medial aspect of the ipsilateral thigh [27].

In the present study, the lymph drainage pattern found for the third neoplastic mammary gland was different from other authors [22,26]. In particular, we observed that one of the third MGT drained into contralateral axillar lymph node, to the authors’ knowledge, this has never been described before. 

In our study, the ipsilateral axillary lymph node and accessory axillary lymph nodes both showed contrast enhancing after draining the third mammary neoplastic gland in one case.

The pattern we have seen concerning the fourth mammary gland is mostly similar to those already described in the literature, with the involvement of the superficial inguinal lymph nodes in six out of seven cases. However, in one case, both superficial inguinal and iliac lymph nodes was observed. In particular, the iliac lymph nodes were never described as lymph drainage pattern for the fourth MGT, while Patsikas it has been demonstrated, in one case of healthy mammary gland lymphatic vessels connecting the mammary gland with the ipsilateral superficial inguinal and medial iliac lymph nodes, simultaneously [31].

The lymph drainage pattern found for the fifth or inguinal neoplastic mammary gland is different in 3 out of 12 cases from other authors [27,30].

Indeed, in these cases, the superficial inguinal and medial iliac lymph nodes were opacified after lipiodol injection in the fifth neoplastic gland. It is postulated that the medial iliac lymph nodes were opacified by lymphatics coming from the fifth MGT and not from the superficial inguinal nodes. In one case, there was no lymph node contrast enhancement and we hypothesized that either the tumor did not find a lymphatic path, or the injections had been done too rapidly and regional lymphatic vessels were damaged.

The results of this study indicate that the lymph drainage pattern of the neoplastic mammary gland may be different in each dog, suggesting that it should be evaluated case by case every time before surgery to really understand the lymph drainage pattern and the SLN as well. 

Indirect lymphography is defined as the deposit of a contrast medium in the periphery of lymphatic vessels in the view of being absorbed and drained by the lymphatic system [20,32].

Indirect lymphography using liposoluble or water-soluble contrast media is a simple method that has been used successfully for the investigation of the lymph drainage in many organs and tissues [31,33,34,35,36,37]. It is considered to reflect the natural lymph flow of the injected area in vivo, demonstrating the efferent lymphatics and the lymph nodes into which they are drained. 

In the present study, indirect lymphography identified the SLN 24 h after injection according to other studies [20,38].

The main advantage the SLN identification by indirect lymphographyis related to the low cost of the procedure. Alternatively, a preoperative lymph node map may be achieved through scintigraphy, but this facility is uncommon in clinical practices [39].

The main limitation of this study is the small number of the study population, and further studies with a bigger number of cases may give more and important information that is still lacking in this field of veterinary medicine. For future perspectives, multicentric studies with lipiodol for mammary or more generally cutaneous and subcutaneous tumors would be of great importance to implement the map of lymphatic drainage in dogs and cats, with the aim of being increasingly precise, both from a therapeutic and prognostic point of view.

## 5. Conclusions

In conclusion, this study provides clinically relevant information with new drainage patterns of the mammary tumors.

The knowledge of the SLN of the neoplastic mammary glands should be of high importance for the surgeon, not only for performing the most adequate surgical excision, but also for determining an accurate post-surgical prognosis.

## Figures and Tables

**Figure 1 animals-11-01115-f001:**
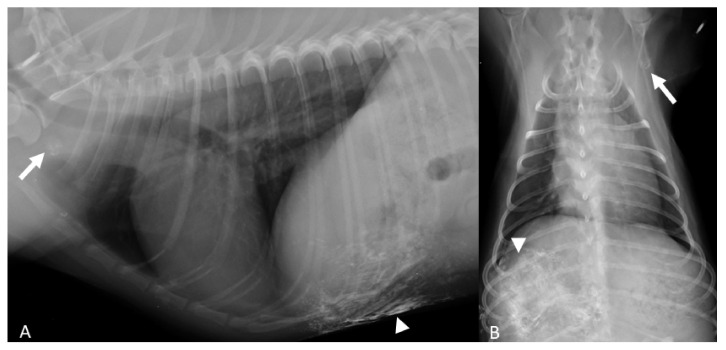
(**A**,**B**) Doberman Pinscher 4 yr (Case #9). Right lateral and ventro-dorsal radiographic projections of the thorax show the lipiodol surrounding the neoplastic mammary gland (arrowheads) and the contrast enhanced contra-lateral axillary lymph node (arrows).

**Figure 2 animals-11-01115-f002:**
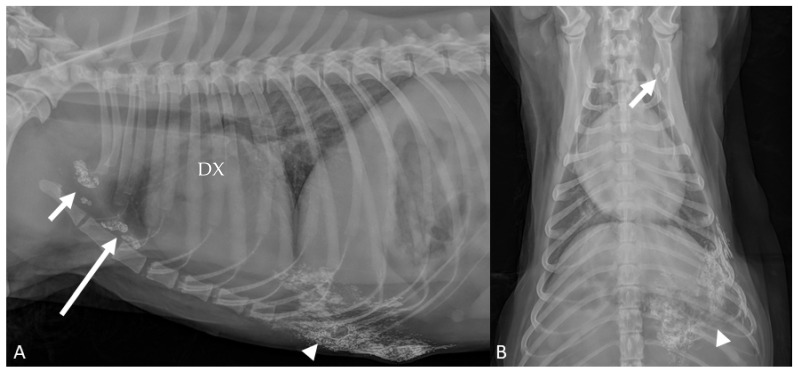
(**A**,**B**) Cocker Spaniel 10 yr (Case #10). Right lateral and ventro-dorsal radiographic projections of the thorax. Third mammary gland surrounded by lipiodol (arrowheads): ipsilateral accessory axillary ((**A**)—long arrow) and a in axillary ((**A**,**B**)—short arrows) lymph nodes were opacified.

**Figure 3 animals-11-01115-f003:**
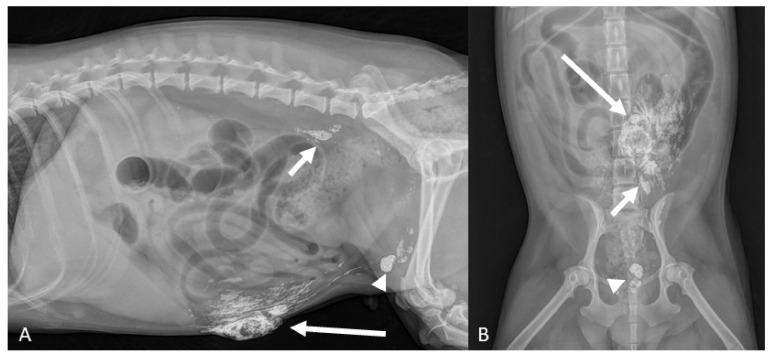
(**A**,**B**) Maltese dog, 8 yr (Case #6). Right lateral and ventro-dorsal radiographic of the abdomen taken 24 h after lipiodol injection in the fourth neoplastic mammary gland ((**A**)—long arrow). Lympatics from the fourth gland that drain into the superficial inguinal lymph nodes are demonstrated ((**A**,**B**)—arrowheads). Medial iliac lymph nodes are also opacified ((**A**,**B**)—short arrows).

**Table 1 animals-11-01115-t001:** Signalament; mammary tumor and lymph drainage; diagnosis and metastatic have been reported; yr: year old; f: entire female; fs: spayed female; ln: lymph node; m: mammary gland.

Signalament	Mammary Gland Tumors Localization and Dimensions (mm)	Lymph Node Drainage	Further Lymph Node Drainage	Histopathology of the Mammary Tumor”	Metastatic Lymph Nodes Involvement
1. German Shepard 6 yr, f, 36 kg.	-left m3 (60 ×d80 mm) -left m5 (50 × 60 mm)	-left axillary ln -left inguinal ln	-left medial iliac ln	-ductal carcinoma I grade	no
2. Mixed breed 8 yr, f, 10 kg.	-left m5 (23 × 25 mm)	-left inguinal ln	-left iliac ln	-ductal carcinoma I grade	no
3. Mixed breed 6 yr, f, 12 kg.	-left m4 (12 × 15 mm) -left m5 (2 × 5 mm)	-left inguinal ln	-left iliac ln	-ductal carcinoma I grade	no
4. Staffordshire 8 yr, f, 28 kg.	-right m4 (4 × 2 × 1 mm)	-right inguinal ln	-n/a	-complex carcinoma I grade	no
5. Labrador retriever, 9 yr, f, 32 kg.	-right m5 (10 × 7.6 mm)	-right inguinal ln	-n/a	-mix carcinoma I grade	no
6. Maltese 8 yr, f, 4 kg.	-left m4 (20 × 35 mm)	-left inguinal ln	-left iliac ln	-micropapillary carcinoma I grade	
7. German Shepard 13 yr, fs, 33 kg.	-right m5 (20 × 20 × 10 mm) -left m4 (5 × 4 × 10 mm)	-right inguinal ln -left inguinal ln	-n/a	-ductal carcinoma II grade right and left.	no
8. Yorkshire 13 yr, f, 7 kg.	-right m4 (10 × 1 mm) -right m5 (12 × 20 mm) -left m4 (10 × 20 mm) -left m5 (20 × 20 mm)	-right inguinal ln -left inguinal ln	-n/a	-right complex adenoma -ductal carcinoma I grade left infection disease	no
9. Doberman Pinscher 4 yr, f, 3 kg.	-right m3 (30 × 40 mm)	-left axillary ln	-n/a	-adenoma	no
10. Cocker Spaniel 10 yr, f, 12 kg.	-left m3 (22 × 25 mm) -left m5 (26 × 4 mm)	-left axillary ln	accessory axillary ln	-fibroadenoma simplex adenoma	no
11. Pitbull terrier 9 yr, f, 22 kg.	-right m5 (20 × 20 mm)	-right inguinal ln		-ductal carcinoma I and III grade	yes
12. Epageul Breton 13 yr, fs, 13 kg.	-left m3 (5 × 5 mm) -left m5 (10 × 20 mm)	-left axillary ln -left inguinal ln		-carcinoma I grade -ductal complex carcinoma II grade	yes
13. Cocker Spaniel 12 yr, f, 12 kg.	-left m4 (10 × 12 mm) -left m5 (20 × 10 mm)	-left inguinal ln		-adenoma	no
14. English Setter 10 yr, f, 16 kg.	-Right m3 (12 × 20 mm) -Right m5 (20 × 20 mm)	-Right inguinal ln		-ductal carcinoma I grade	no

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
