# Peer review of "Lymphatic Drainage Mapping with Indirect Lymphography for Canine Mammary Tumors"

_animals, 2021, doi:10.3390/ani11041115_

Round 1

Reviewer 1 Report

animals-1166269-peer-review-v1

General comments: The authors have mapped lymph node drainage associated with mammary neoplasia in dogs. While many of the findings are expected, new findings of additional draining lymph nodes associated with distinct mammary glands have been reported. A few general comments:

1) In the abstract and elsewhere it is discussed that distant metastasis should not be present if the SLN does not have evidence of tumor. This is true with lymphatic metastasis, but NOT with hematogenous metastasis, which is also another source. These passages should be changed to reflect this.

2) The sections of the manuscript need less paragraphs. Many of these small paragraphs should be combined into larger paragraphs.

3) Table 1 indicates other mammary glands have disease within the same dog. The methods section should be edited to indicate multiple glands were evaluated per dog.

More specific comments are listed below by section and line number:

Methods

1) Line 110 – This last sentence of this paragraph should be reworded.

Author Response

Dear Reviewer,

thank you very much fr your comments and suggestions.

Please find attached the rebuttal letter.

Kind regards

Reviewer 2 Report

The manuscript "Lymphatic Drainage Mapping with Indirect Lymphography for Canine Mammary Tumors" is quite interesting and well written. However, there are some points to improve.

Introduction.

  • references considered in lines 33-35 must be updated, especially with regard to the probability of malignancy. There are references that report different incidences of malignancy 
  • line 75 is not necessary to include. I suggest to erase or modify
  • the aim of this study is very general and ambiguous. Please, modify to generate a clear conclusion

Methods

  • The "n" studied should be included

Discussion

  • I suggest including some future perspectives from this study

Author Response

(The authors gave the same response as above.)

Reviewer 3 Report

Dear Colleagues, thank you for submitting this interesting paper to us. The subject dealt with is certainly of practical interest. I kindly ask you to give me some clarification on the matter. In the materials and methods section in line 107 only the clinical diagnosis of neoplasm is reported, why did you not consider taking samples for cytological examination?
Furthermore, in line 105 in the description of the technique it is not indicated at what distance from the neoplasm the injection is carried out.
Why did you fail to include lesions of the first and second mammary gland in the series? In the RV radiographs of case # 9 it is noted how the contrast medium reaches the midline, could this be the reason why the contralateral axillary lymph node was involved?
In line 161 did you explain why there was no collection? I believe from table 1 that this is case # 12 but it should be better reported in the table.
In table 1 to the description of case # 8 there is no alignment between neoplasms and lymph nodes (right / left).
Thanks again, I offer you my best regards.

Author Response

(The authors gave the same response as above.)

Round 2

Reviewer 1 Report

animals-1166269 - R1

General Comments: The authors have nicely addressed all of my comments. One minor point:

  • Line 229 – Change “hypnotized” to “hypothesized”

Reviewer 3 Report

Dear Authors, thank you for having promptly applied the changes requested to the paper in question. Yours sincerely